# Dynamics and resiliency of networks with concurrent cascading failure and self-healing

**Waseem Al-Aqqad[1], Hassan S. Hayajneh[2], Xuewei Zhang[1]** *

**1** Department of Electrical Engineering and Computer Science, Texas A&M University-Kingsville, Kingsville, Texas, United States of America, **2** Department of Engineering Technology, Purdue University Northwest, Hammond, Indiana, United States of America

* xuewei.zhang@tamuk.edu

## Abstract

Local attacks in networked systems can often propagate and trigger cascading failures. Designing effective healing mechanisms to counter cascading failures is critical to enhance system resiliency. This work proposes a self-healing algorithm for networks undergoing load-based cascading failure. To advance understanding of the dynamics of networks with concurrent cascading failure and self-healing, a general discrete-time simulation framework is developed, and the resiliency is evaluated using two metrics, i.e., the system impact and the recovery time. This work further explores the effects of the multiple model parameters on the resiliency metrics. It is found that two parameters (reactivated node load parameter and node healing certainty level) span a phase plane for network dynamics where three regimes exist. To ensure full network recovery, the two parameters need to be moderate. This work lays the foundation for subsequent studies on optimization of model parameters to maximize resiliency, which will have implications to many real-world scenarios.

**Data Availability Statement:** The data underlying the results presented in the study are available from: https://figshare.com/articles/software/Final_m/21453858.

## Introduction

A wide range of real-world systems such as power grids [1], financial transaction networks [2], communication networks (e.g., the Internet) [3], and command and control systems [4] have been modeled as complex networks. Among other characteristics, the resiliency of networked systems has received growing research attention from diverse application areas including economic systems [5], organizational management [6], and multiple engineering systems [7, 8]. Generally speaking, resiliency can be viewed as the ability of a system to bounce back from high-impact disruptions to achieve partial or full recovery [9]. To improve the resiliency of a system, it is necessary to advance the knowledge of the effects of system properties, external disruptions, and recovery mechanisms on resiliency, which calls for extensive modeling and simulation studies and would be of fundamental interests to the planning, design, operation, and control of systems from critical infrastructure and supply chains to disaster recovery and humanitarian aids [10].

**Funding:** This work received support from the U.S. Department of Commerce, Economic Development Administration (https://eda.gov/) under Award #08-69-05349 of which X.Z. is the principal investigator. There was no additional funding received for this study.

**Competing interests:** The authors have declared that no competing interests exist.

## Review of related works

To characterize system resiliency, the modeling and simulation efforts need to consider two aspects: (1) following an initial (usually local) attack, the failure of nodes or links propagates over the network, which is called cascading failure; (2) at certain point the system's self-healing mechanism is activated, which would counter the impacts of cascading failure. In addition, it is also necessary to develop consistent, quantitative resiliency metrics to facilitate the comparison of system performances.

Previous research on cascading failure in networks (here we focus on single-layer networks) fall into two main categories: connectivity-based [11, 12] and load-based [13, 14]. The first cascading failure model was developed in [11] to describe the propagation of binary decisions in a population of interacting decision-makers (nodes). Each node observes the states (0 or 1) of the nodes connected to it (neighbors). Its state to be in state 1 (active) or state 0 (inactive) is determined from whether the fraction of its neighbors being in state 1 is higher or lower than a pre-specified threshold. It was found that large cascades can be triggered due to the inactivation of highly connected nodes. In [12], it was shown that community structures are crucial in connectivity-based cascading failures such as information diffusion and virus spreading. In [13], the nodes in a network were associated with a physical quantity called "load" that can be transferred between neighbors, and the effects of inactivating some nodes with their loads transferred to the neighboring nodes were investigated. It was assumed that the initial load of a node is the total number of shortest paths passing through the node. If the load of a node exceeds its capacity (which is, by definition, proportional to its initial load), the node will become inactive, and its load will be transferred to its neighbors. It was shown in [14] that the networks with more heterogeneous distribution of loads are likely to be more vulnerable to cascades of overload failures.

These two types of cascading failure (connectivity-based and load-based) can be used to model many systems under disruptions. However, the majority of prior studies have focused on network robustness (i.e., how to mitigate the risk of global failure), without considering active defense or self-healing (recovery) processes that could be initiated after some damages have been made to original system. Self-healing in complex networks has raised substantial research interests in the past decade. Representative studies on network self-healing can be found in [15–18]. For instance, a defense strategy against cascading failure due to overload was proposed in [15], which was based on selective removal strategy of nodes/links immediately after the initial attack. It was shown that the removal of nodes with low loads can result in reduced size of cascades. Two self-healing models were introduced in [16, 17], where the former decides for each node, after damage, whether to create a new link depending on the fraction of neighbors it has lost, while the implementation of the latter relies on the presence of dormant backup links that can be switched back on. However, these studies developed solutions to repair or restore system "instantaneously" and did not treat network recovery as a dynamic process [18]. As such, self-healing in load-based failure scenarios has not been modeled [19]. Further, there have been few studies on a networked system with concurrent cascading failure and healing [20].

There have been some studies evaluating the resiliency of complex networks. In [7], some metrics were designed to quantify the resiliency of networked infrastructure systems during earthquakes and hurricanes. An agent-based modeling approach was demonstrated in [9] to assess the performance of a complex system after disruptions using metrics such as systemic impact and the time to reach a full restoration. A method called resilience triangle was introduced in [21] to quantitatively assess three aspects of supply chain network resiliency:

complexity, density, and node-criticality. Another concept called expected disruption cost was proposed in [22] to quantify resiliency and enable its inclusion in optimization models.

### Objective and overview of this work

This work aims to consider networked systems with concurrent load-based cascading failure and self-healing and investigate the dependence of resiliency on various system parameters. The dynamic healing model for overload failures is newly developed. The effects of some important healing parameters such as triggering level and budget parameter were explored. Two metrics, i.e., 95% recovery time and T-20 active node number, are used to measure the resiliency. The networks under study in this paper include an Erdös-Rényi (ER) random network [23] and a scale-free (SF) network [24].

The major contribution of this work is to develop a dynamic modeling and simulation framework to quantitatively assess the resiliency of networked systems. To the best of our knowledge, this is the first time when cascading failure, self-healing, and resiliency are considered together as integral parts of a dynamic networked system. This dynamic system modeling framework also enables the examination of the effects of the model parameters on resiliency. This work lays the foundation for subsequent studies on more complex mechanisms and processes on the networks, optimization of resiliency, as well as applications to more real-world scenarios.

The organization of the remainder of this paper is as follows. Sec. 2 describes the load-based cascading failure and self-healing models implemented in this work, as well as the resiliency metrics. Sec. 3 presents the ER and SF networks under study, the system dynamics (i.e., recovery trajectories) under various combinations of parameters, and the corresponding resiliency metrics. The conclusions of the paper are drawn in Sec. 4.

## Model and methods

### Dynamic processes on networks

In this work, the systems under study are networks (or graphs). The initial network is denoted as $G$, while the network at each time step following the initial attack is denoted as $G\_dmg$. Table 1 lists the most important notations used in the system model. The system model consists of four main modules, each of which is described below.

(i) Initialization ($t = 0$). This module has two parts. Firstly, since the processes on the network are load based, it is needed to specify the initial load $L_{i,0}$ and capacity (maximum load) $C_i$ of each node $i = 1, 2, \ldots, N$. In this work, the $L_{i,0}$'s are sampled from uniform distribution over $[L_{min}, L_{min}]$. The capacity of each node is assumed to be proportional to its initial load, i.e., $C_i = (1 + a)L_{i,0}$ where $a$ is called tolerance factor (fixed at 0.1 throughout this work). The $C_i$'s, once set, will remain constant in the simulation. Secondly, in the original network $G$, a set (denoted $R$) of nodes are selected as the targets of initial attack (each with an additional load shock $D$). In this work, the nodes in $R$ are randomly selected from all nodes in $G$.

(ii) Cascading ($t = 1, 2, \ldots, t_{max}$). At each time step $t$, consider the network $G\_dmg$ from the previous time step $t − 1$. For $i = 1, \ldots, N$, if $L_{i,t−1} > C_i$, then node $i$ fails and becomes inactive (in our work a load of negative infinity is assigned to inactive nodes), and the load $L_{i,t−1}$ is transferred uniformly to the active neighbors of node $i$. The transfer process starts with the identification of the set of neighboring nodes of node $i$ in $G$, denoted by $b_i$, and then filter out the elements in $b_i$ with a negative infinity load to obtain the set of active neighboring nodes of node $i$ in $G\_dmg$, denoted as $Active\_b_i$. The load of each node in $Active\_b_i$ will be increased by $L_{i,t−1}/\text{length}(Active\_b_i)$. The resulting network is returned as the updated $G\_dmg$.

**Table 1. List of notations.**

| Symbol | Description |
|---|---|
| $G$ | Initial network |
| $G\_dmg$ | Damaged network |
| $N$ | Number of nodes in $G$ |
| $t$ | (Discrete) time step |
| $t\_max$ | Maximum time steps of simulation |
| $t\_trig$ | Time step of self-heating being triggered |
| $i, j$ | Index of node in the networks |
| $L_{i,t}$ | Load of node $i$ at time $t$ |
| $L_{i,0}$ | Initial load of node $i$ |
| $C_i$ | Capacity of node $i$ |
| $L_{min}$ | Lower bound of initial load distribution (uniform) |
| $L_{max}$ | Upper bound of initial load distribution (uniform) |
| $a$ | Tolerance factor |
| $R$ | Set of initially attacked nodes |
| $D$ | Shock added to the load of a node under initial attack |
| $b_i$ | Set of neighboring nodes of node $i$ |
| $Active\_b_i$ | Set of active neighboring nodes of node $i$ |
| $T$ | Triggering level for self-healing, normalized by $N$ |
| $B$ | Budget parameter of self-healing, normalized by $N$ |
| $LC\_Ratio_i$ | Mean of capacity usages of inactive node $i$'s active neighbors |
| $\alpha$ | Certainty level of node healing |
| $P$ | Portion of active neighbors' mean load transferred to a reactivated node |
| $A_t$ | Number of active nodes at time step $t$, normalized by $N$ |
| $T_\beta$ | Time steps needed to reach $\beta$% recovery |

(iii) Healing ($t = t_{trig}, \ldots, t_{max}$). If the number of inactive (failed) nodes has not reached a pre-specified threshold $T$, the self-healing will not be initiated. Once the number of inactive nodes exceeds $T$ at time step $t_{trig}$, the self-healing module will be running. In the model, it is possible to implement $x$ repetitions of module (ii) and $y$ repetitions of module (iii) at each time step to model the different "speeds" of the two processes; however, in this work, we set both $x$ and $y$ to be 1. The healing process has two steps as follows.

Step 1—Decision. By selecting some inactive nodes to recover (reactivate) at each time step, we aim to maximally mitigate the cascading failure and restore the original network as quickly as possible. The highest number of nodes that can be recovered at each time step is called the budget parameter of healing, $B$. Our model always attempts to recover the highest possible number of inactive nodes; when the number of inactive nodes is greater than $B$, we need to rank these inactive nodes by their "importance". In this work, the importance is approximately evaluated as the average capacity usage of the inactive node's active neighbors. For each inactive node $j$ in $G\_dmg$, one can identify $Active\_b_j$ and calculate the mean of the ratios of current load and capacity of all nodes in $Active\_b_j$. The result is denoted by $LC\_Ratio_j$; the inactive nodes with higher $LC_{ratio}$ will be of higher priorities to be chosen. It makes sense to immediately restore the inactive nodes whose active neighbors have the highest average capacity and are the most vulnerable to failures in the next time step. These highest impact inactive nodes (limited by the set size and $B$) will be the input of Step 2 below.

Step 2—Implementation. This step is the reactivation of the inactive nodes identified above. For each of such inactive node $j$, we will transfer some of the load of each of its active neighbors

to itself (for the relief of the active neighbors). In our model, each neighbor transfers a portion $P$ of the mean of the loads of these active neighbors. The load of node $j$ is updated from negative infinity to the transferred amount. The load of each of those neighbors will be reduced by $P/\text{length}(Active\_b_j)$ of its old value. Furthermore, another parameter is introduced, namely, certainty level of node healing $\alpha \in [0, 1]$. This parameter captures the success probability of the implementation of healing of any nodes selected in Step 1. The output of Step 2 will be the updated $G\_dmg$.

(iv) Resiliency evaluation. The algorithm outlined above can generate system trajectories (number of inactive nodes vs time), based on which one can evaluate resiliency. In this work, we examine two metrics: $A_t$, the number of active nodes (normalized by $N$) at time step $t$, and $T_\beta$, time steps needed to reach $\beta$% recovery (i.e., the network reaches a steady state with less than $(100 - \beta)$% inactive nodes). Higher values of $A_t$ (less severe disruption) and lower values $T_\beta$ (faster recovery) correspond to more resilient systems.

## Research design

In this work, we consider two different setups of the initial network $G$: (1) a computer generated ER network and (2) a computer generated SF network. Both networks have 5000 nodes ($N = 5000$) connected by 10000 links. For the ER network, we randomly generate the adjacency matrix and pick one with all nodes connected. For the SF network, we start from 5 interconnected nodes (seed) and add new nodes. The number of links a new node can make to the existing nodes is 2. This repeats until the total node number reaches $N$.

The four model parameters to be investigated are: $T$, $B$, $P$, and $\alpha$, all between 0 and 1. For various combinations of these parameters, we will compare the resulting system trajectories and resiliency metrics to explore the effects of each parameter.

## Results and discussion

### System trajectories

Fig 1 presents the system trajectories for two different network topologies (ER and SF). The initial attacks are targeted at 8 nodes that are randomly chosen in the simulation. As expected,

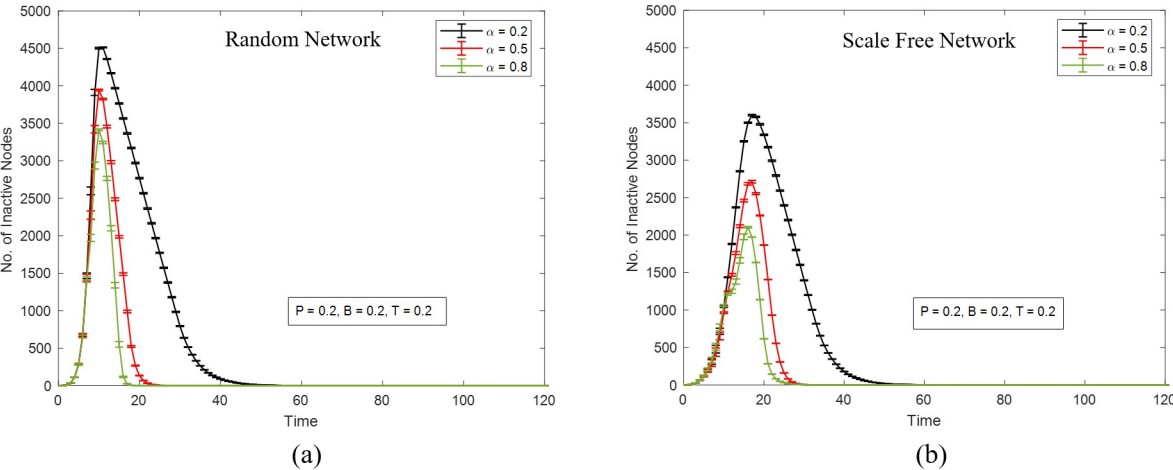

(a)                                                                                      (b)

**Fig 1. The number of inactive nodes as a function of time steps in (a) a random network and (b) a scale-free network.** Both networks have 5000 nodes and 10000 links. $T = B = P = 0.2$, and $\alpha$ takes three values: 0.2 (black curve), 0.5 (red curve), and 0.8 (green curve). Each data point is the average of results of 20 iterations of simulation (each with the same initial attack targets). The error bars indicate standard errors.

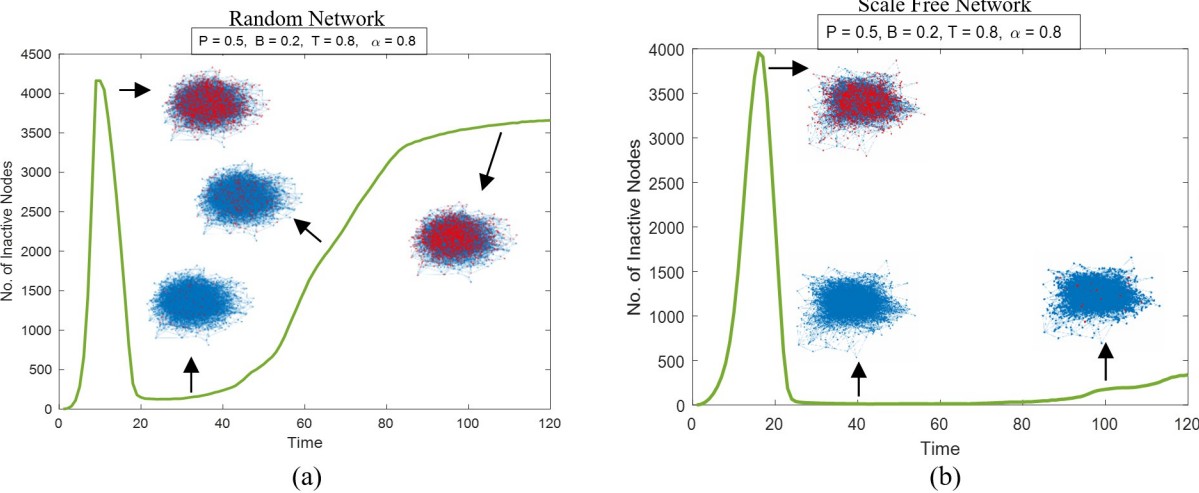

**Fig 2. The number of inactive nodes as a function of time steps at higher values of *P* and *α* in (a) a random network and (b) a scale-free network.** Both networks have 5000 nodes and 10000 edges. The snapshots indicate the distributions of the inactive nodes (red) and the active nodes (blue) at respective time steps.

the self-healing mechanism works, since the number of inactive nodes in all cases eventually falls to zero (or very close to zero). In both networks, under the current parameter settings, a higher certainty level *α* leads to faster recovery and smaller damage, which agrees with the intuition that higher certainty of node healing means higher recovery efficiency. Additionally, the SF network appears to be more resistant to random attacks, as the peak number of inactivated nodes are lower than that of the ER network. However, the recovery in the SF network takes longer than that in the ER network. This is a consequence of our healing mechanism. When the damage is more severe at the time of healing initiation (and the budget parameter is high enough), at subsequent time steps there will be more inactive nodes of high importance to be selected for reactivation, which in general results in a faster system recovery.

Higher certainty levels *α* could produce counter-intuitive outcomes, especially as *P* is increased. Fig 2 shows trajectories of the two networks in Fig 1 with *α* = 0.8, *P* = 0.5, *B* = 0.2, and *T* = 0.8, as well as snapshots of the damaged networks at different time steps. In both cases, the system cannot reach a full recovery. The self-healing process works well initially, bringing the number of inactive nodes down to >90% recovery. After that, the cascading failure regains momentum. And the number of inactive nodes grows more rapidly in the ER network than in the SF network, which is consistent with the observed trend of cascading failure in Fig 1 (note that it is out of the scope of this work to examine whether this holds for all possible pairs of ER and SF networks). The "re-ignition" of cascading failure is seeded by the reactivated nodes with transferred loads that are higher than their capacities (defined at *t* = 0 as (1 + *a*) times initial loads and kept constant) when *P* is high. If the capacity is updated as (1 + *a*) times the new load, the phenomenon seen in Fig 2 would disappear. We stick with the original model in this work, in view of that in many real-world cases, when restoring system components, previous specifications are often followed.

Combining the results in Figs 1 and 2, it is implied that lowering the certainty level *α* could offset some negative impacts of increased *P*, by reducing the occurrences of the reactivated, overloaded nodes. This can be seen more clearly in Fig 3, where different combinations of *P* and *α* correspond to different behaviors of the ER network. In S1, both *P* and *α* are too small to generate healing strong enough to counter the cascading failure. S2 is a region with either or

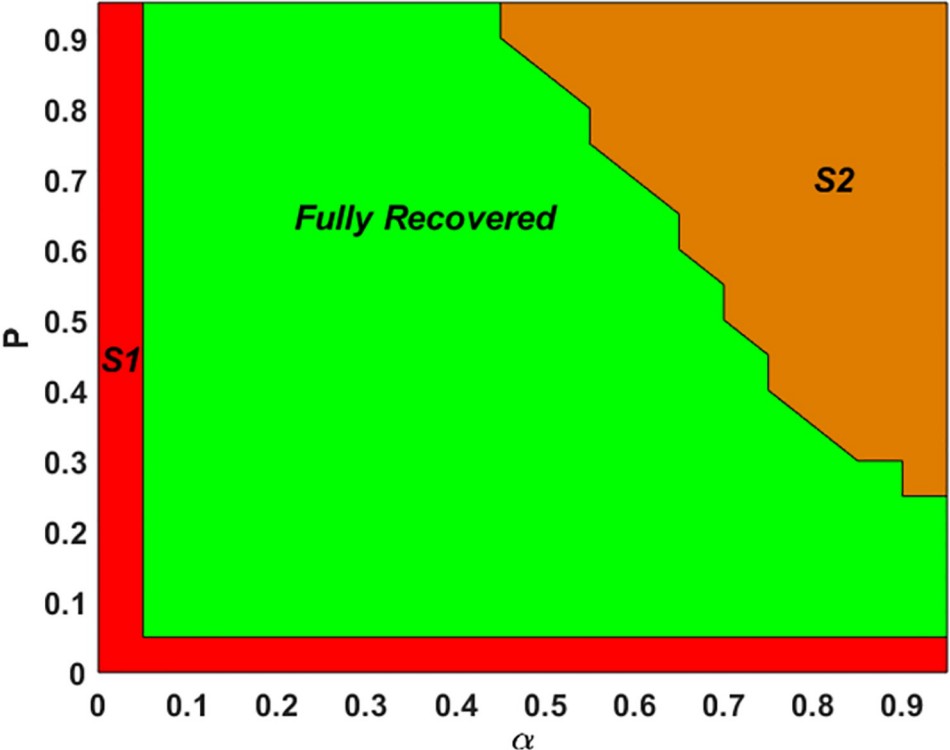

**Fig 3. Phase diagram of the ER network behavior under various combinations of *P* and *α* (both from 0 to 1 at 0.05 step size) with *T* = 0.2 and *B* = 0.8.** There are three regimes: S1 (red)—healing unable to stop cascading failure; Fully Recovered (green); and S2 (brown)—healing unable to achieve 100% (or very close to 100%) recovery.

both of *P* and *α* being high, where the long-term behavior of the system is either partially recovered at a stable level or oscillating around a certain level. The zigzagged boundary between S2 and the fully recovered regime does exhibit a trend of decreasing *α* with increasing *P*.

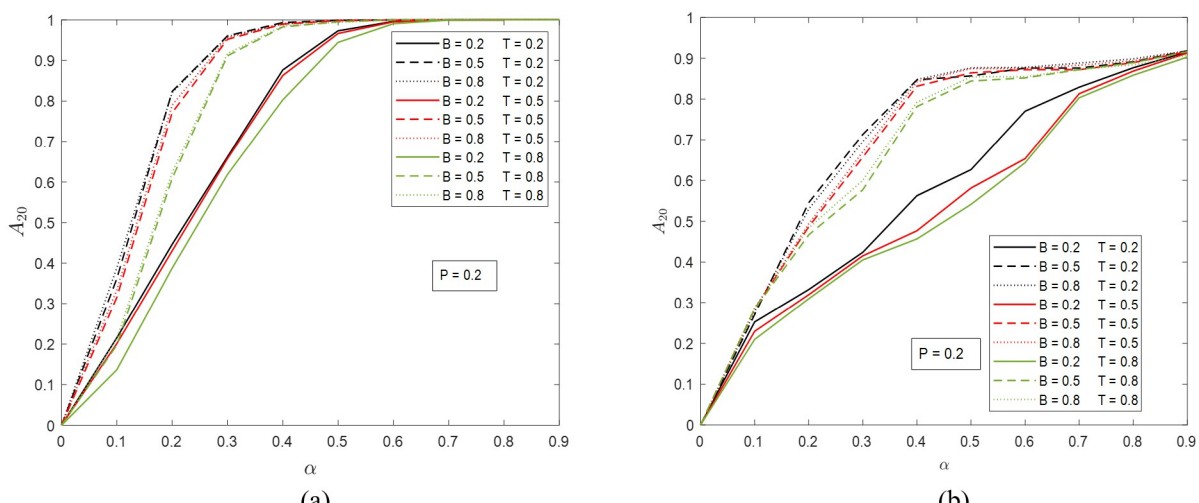

(a)　　　　　　　　　　　　　　　　(b)

**Fig 4. The portion of active nodes at time step 20 when *P* = 0.2 in (a) the ER network and (b) the SF network.** The possible values of *B* and *T* are 0.2, 0.5, and 0.8. The values of *α* are from 0 to 0.9 at step size 0.1.

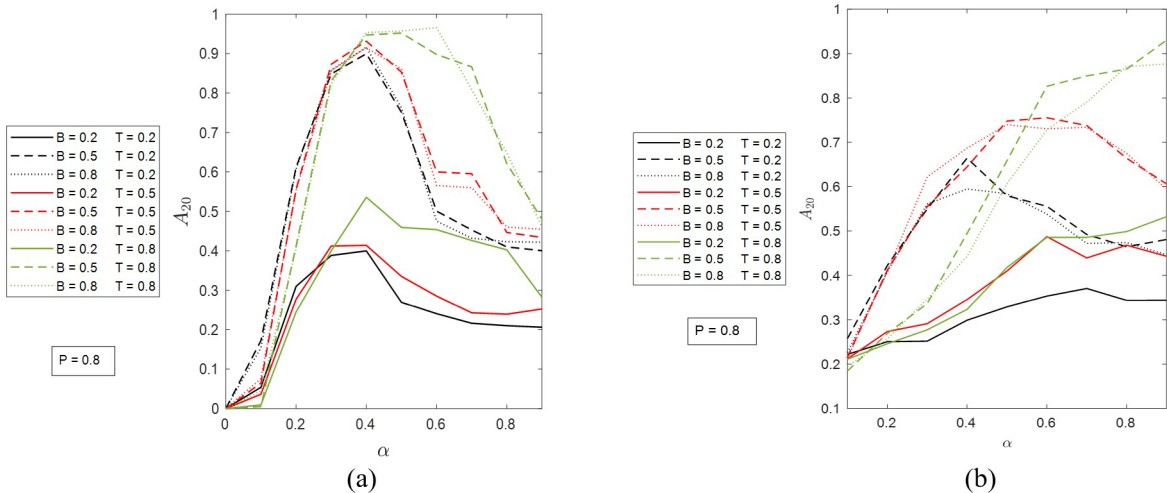

**Fig 5. The portion of active nodes at time step 20 when P = 0.8 in (a) the ER network and (b) the SF network.** The possible values of $B$ and $T$ are 0.2, 0.5, and 0.8. The values of $\alpha$ are from 0 to 0.9 at step size 0.1.

**Resilience metrics.** We now move on to measure the resiliency of the networks and investigate the effects of the four model parameters. Here we consider two metrics: (a) $A_{20}$, the number of active nodes (normalized by $N$) at time step $t = 20$; and (b) $T_{95}$, the time needed to reach 95% recovery (i.e., <5% inactive nodes). In Figs 4 and 5, we show the results of $A_{20}$ for the ER and SF networks with different $B$, $T$, $P$, and $\alpha$ values. As shown in Fig 4, when $P = 0.2$, for both networks, a higher $T$ (the same $B$) always corresponds to a lower $A_{20}$, which means that the later the self-healing mechanism kicks in, the more severe the damage observed at $t = 20$, and the less resilient the system. The difference between the two networks is the effect of $\alpha$. In the ER network, when $\alpha$ is greater than 0.6, the $A_{20}$ values of all cases approach 1, while in the SF network, the convergence is not as obvious. Comparing the results with different $B$ values under the same $T$, one can see that as $B$ increases beyond certain level, no

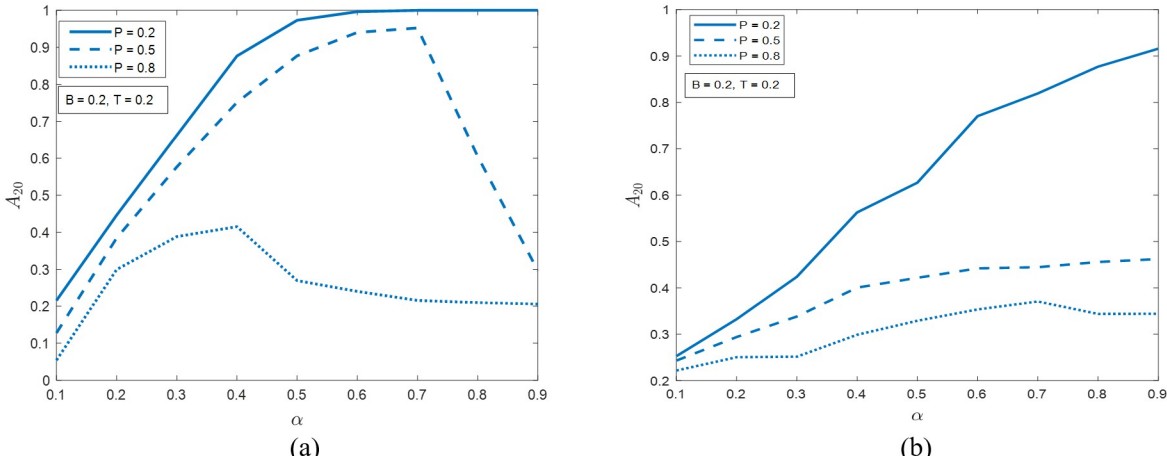

**Fig 6. The portion of active nodes at time step 20 when B = T = 0.2 in (a) the ER network and (b) the SF network.** The possible values of $P$ are 0.2, 0.5, and 0.8. The values of $\alpha$ are from 0 to 0.9 at step size 0.1.

significant increase in $A_{20}$ will be made. For instance, in Fig 4(a), the results with $B = 0.5$ and $B = 0.8$ (the same $T$) are very close to each other. The apparently larger deviations seen in Fig 4 (b) are most likely a result of the randomness in the simulations.

In Fig 5(a), when $P = 0.8$ and $\alpha < 0.4$, the dependence of $A_{20}$ of the ER network on $T$ and $B$ are the same as in Fig 4(a). Under higher $\alpha$ values, because of the re-ignition of cascading failure shown in Fig 2(a), the expected effect of $T$ on $A_{20}$ (the same $B$) is no longer seen. However, a common feature of all curves is that $A_{20}$ reaches maximum at $\alpha$ 0.4. The results in Fig 5(b) for the SF network are more chaotic. The dependence of $A_{20}$ on $B$ (the same $T$) remain the same as in Fig 4(a). The budget "saturation" effect comes from the healing algorithm and is universal in all simulation cases.

Fig 6 demonstrates the effects of $P$ and $\alpha$ on the resiliency metric $A_{20}$. In both networks, with $P$ increasing from 0.2 to 0.8, at all $\alpha$ values, $A_{20}$ will decrease, which means that the

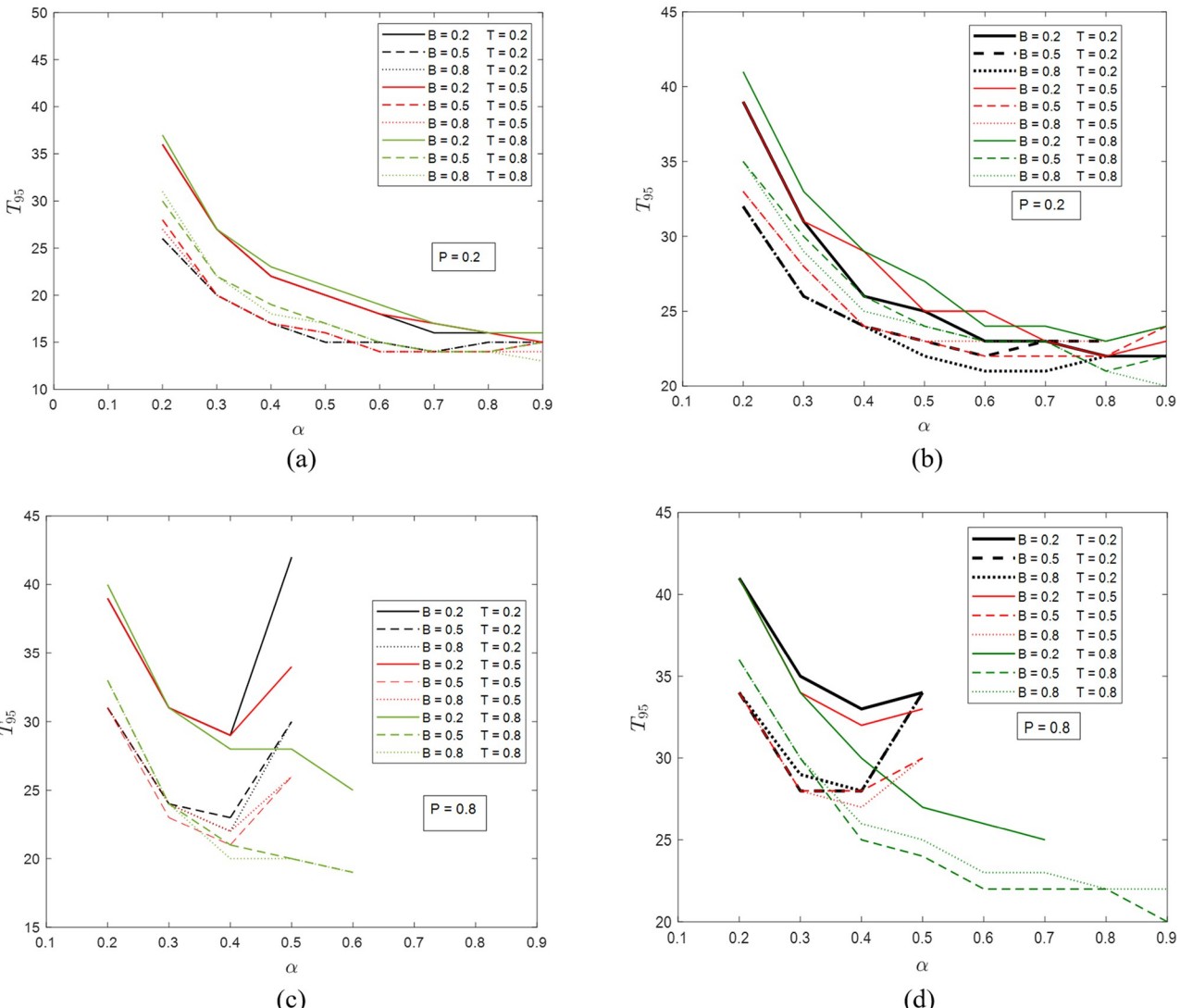

**Fig 7. The time needed for 95% recovery in (a) the ER network with $P = 0.2$, (b) the SF network with $P = 0.2$, (c) the ER network with $P = 0.8$, and (d) the SF network with $P = 0.8$.** The possible values of $B$ and $T$ are 0.2, 0.5, and 0.8. The values of $\alpha$ are from 0.2 to 0.9 at step size 0.1. In some cases with higher $\alpha$, the system can never reach 95% recovery and therefore no data points are shown in the figures.

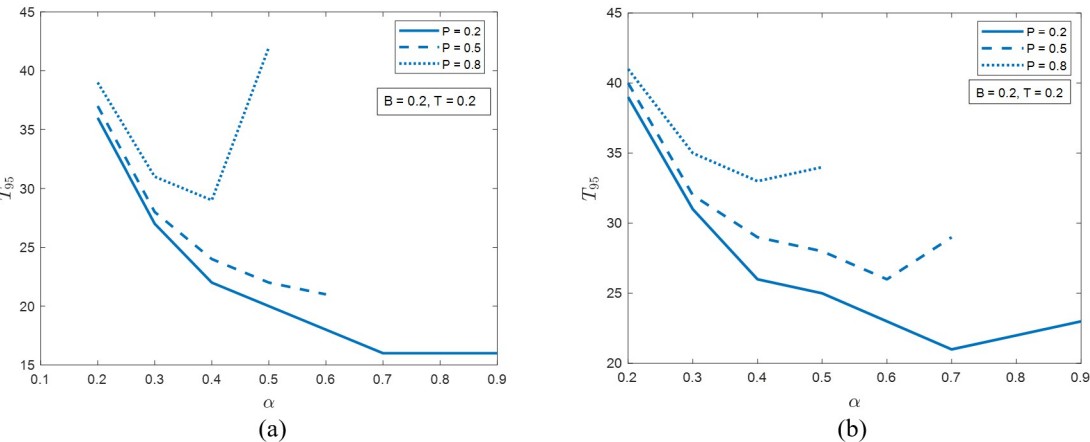

**Fig 8. The time needed for 95% recovery in (a) the ER network and (b) the SF network with $B = T = 0.2$.**

systems become less resilient. For smaller $P$ (e.g., 0.2), $A_{20}$ tends to increase with increasing $\alpha$, while for larger $P$, $A_{20}$ is more likely to peak at an intermediate $\alpha$.

The use of $A_t$ as resiliency metric has its limitation because it contains only the information of a snapshot of system dynamics. Moreover, the assessment of system resiliency based on $A_t$ might not yield consistent conclusions simply because of the selection of different observation times. In view of this, we measure the resiliency using another metric called recovery time $T_\beta$ ($\beta = 95$ in this work). This metric, compared to the prior one, can fully capture the overall system dynamics (it is also worth noting that while $T_\beta$ is a more comprehensive and consistent metric for resiliency planning, $A_t$ is mostly used for real-time decision-making).

In Fig 7, $T_{95}$ results of the two networks under various combinations of parameters are presented. The intuition is that, with the increase of node healing certainty level $\alpha$, $T_{95}$ decreases (i.e., faster recovery and better resiliency). However, this is only true when the system is in the Fully Recovery regime (Fig 3), i.e., neither $P$ nor $\alpha$ can be very high, which is supported by the results in Fig 7. One can observe a converging trend in $T_{95}$ values with increasing budget parameter $B$ and other parameters the same, which is the same as the aforementioned budget saturation effect. The effects of triggering level $T$ appear to be entangled with other parameters and cannot be easily separated.

Fig 8 shows the dependence of $T_{95}$ on $P$ and $\alpha$ in the two networks with $B = T = 0.2$. In both systems, the common counterintuitive result is that the increase of $P$ from 0.2 to 0.5 and then to 0.8 cannot reduce the recovery time; actually it does the opposite. This trend of system resiliency in Fig 8 is consistent with that in Fig 6, showing the potential of compatibility of the two metrics.

## Conclusions

In this work, we propose a self-healing mechanism for networks undergoing load-based cascading failure. We develop a simulation framework to study the resiliency of networked systems with concurrent cascading failure and self-healing. The two resiliency metrics used are the time-$t$ active node portion ($A_t$) and the time for $\beta$% recovery ($T_\beta$). The network resiliency has the following dependencies on the model parameters.

1. Budget parameter $B$. If it is too small, the healing process is able to counter cascading failure. When $B$ is high enough, further increasing it cannot bring additional improvements in resiliency (budget saturation effect).

2. Reactivated node load parameter $P$ and node healing certainty level $\alpha$. As illustrated in Fig 3, if either is very small, the network cannot have effective healing; when either is high, the system might enter a regime where no full recovery can be made. To make sure that full recovery occurs, our model requires the specification of moderate $P$ and $\alpha$. And it is possible to find the combinations of $P$ and $\alpha$ that maximize the resiliency.

3. Triggering level $T$. In cases with moderate $P$, $\alpha$, and sufficiently high $B$, lowering $T$ (sooner healing kick-in) could increase $A_{20}$ (but not necessarily reduce $T_{95}$). In general, the effect of this parameter is highly entangled with other parameters.

In addition, this work provides preliminary results showing the difference between the ER and SF networks in terms of system trajectories and resiliency metrics. We also see the promise of making the two resiliency metrics consistent by specifying appropriate $t$ and $\beta$.

This work lays the foundation for subsequent studies on more complex mechanisms and processes on the networks, optimization of parameters to maximize resiliency, and applications to more real-world scenarios.

## Author Contributions

**Conceptualization:** Waseem Al-Aqqad, Xuewei Zhang.

**Data curation:** Waseem Al-Aqqad.

**Formal analysis:** Waseem Al-Aqqad.

**Funding acquisition:** Xuewei Zhang.

**Investigation:** Waseem Al-Aqqad, Xuewei Zhang.

**Methodology:** Waseem Al-Aqqad, Xuewei Zhang.

**Project administration:** Hassan S. Hayajneh, Xuewei Zhang.

**Resources:** Hassan S. Hayajneh.

**Software:** Waseem Al-Aqqad, Xuewei Zhang.

**Supervision:** Xuewei Zhang.

**Validation:** Waseem Al-Aqqad, Hassan S. Hayajneh, Xuewei Zhang.

**Visualization:** Waseem Al-Aqqad.

**Writing – original draft:** Waseem Al-Aqqad, Hassan S. Hayajneh, Xuewei Zhang.

**Writing – review & editing:** Waseem Al-Aqqad, Hassan S. Hayajneh, Xuewei Zhang.

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
