## [Decision Letter · Decision Letter 0]

22 Aug 2022

PONE-D-22-20047Dynamics and resiliency of networks with concurrent cascading failure and self-healingPLOS ONE

Dear Dr. Zhang,

Thank you for submitting your manuscript to PLOS ONE. After careful consideration, we feel that it has merit but does not fully meet PLOS ONE’s publication criteria as it currently stands. Therefore, we invite you to submit a revised version of the manuscript that addresses the points raised during the review process.

We look forward to receiving your revised manuscript.

Kind regards,

Jiashen Teh

Academic Editor

PLOS ONE

Journal Requirements:

"This work is supported in part by the U.S. Department of Commerce, Economic Development Administration (https://eda.gov/) under Award #08-69-05349 of which X.Z. is the principal investigator."

Additional Editor Comments:

The reviewers have expressed some concerns pending final acceptance. Please revised the paper according to the comments provided

Reviewers' comments:

Reviewer's Responses to Questions

**Comments to the Author**

1. Is the manuscript technically sound, and do the data support the conclusions?

Reviewer #1: Yes

Reviewer #2: Yes

2. Has the statistical analysis been performed appropriately and rigorously? 

Reviewer #1: Yes

Reviewer #2: N/A

3. Have the authors made all data underlying the findings in their manuscript fully available?

Reviewer #1: Yes

Reviewer #2: No

4. Is the manuscript presented in an intelligible fashion and written in standard English?

Reviewer #1: Yes

Reviewer #2: No

5. Review Comments to the Author

Reviewer #1: This work proposes a self-healing algorithm for networks undergoing load-based cascading failure. To advance understanding of the dynamics of networks with concurrent cascading failure and self-healing, a general discrete-time simulation framework is developed, and the resiliency is evaluated using two metrics. This work further explores the effects of the multiple model parameters on the resiliency metrics. I think the work merits a publication in this journal, but I have the following concerns:

1)) The DTR system has been shown in [“Comprehensive review of the dynamic thermal rating system for sustainable electrical power systems”, Energy Reports] to have significant benefits towards the rating enhancement of the transmission networks. It enables flexible rating to withstand contingencies and improve network reliability in general, as shown in

(i) [“Impact of the real-time thermal loading on the bulk electric system reliability”, IEEE Trans Reliability],

(ii) [“Uncertainty analysis of transmission line end-of-life failure model for bulk electric system reliability studies”, IEEE Trans Reliability],

(iii) [“Network topology optimisation based on dynamic thermal rating and battery storage systems for improved wind penetration and reliability”, Applied Energy] and

(iv) [“Reliability impacts of the dynamic thermal rating and battery energy storage systems on wind-integrated power networks”, SEGAN] before.

Hence, it is clear that the DTR system can benefit the self-healing effort proposed here by providing more headrooms to the line rating. However, the DTR system is never considered in the proposed model. Due to this, it is suggested that the authors additionally consider the DTR system in their proposed model, otherwise, they should provide a qualitative discussion on how the DTR system could have been included and its potential impacts towards the presented results.

2)) The analysis of the proposed methodology has focused only on the physical dimension of the power system and ignores the cyber layer. Studies such as in [“Reliability impacts of the dynamic thermal rating system on smart grids considering wireless communications”, IEEE Access] and [“Surveys on the reliability impacts of power system cyber–physical layers”, Sustainable Cities and Societies] have shown that the cyber layer exerts significant reliability threat to the power system and it should not be ignored. In terms of the DTR system, such a modelling have also been undertaken before in [“Impacts of Communication Network Availability on Synchrophasor-Based DTR and SIPS Reliability”, IEEE Systems Journal] and [“Composite Reliability Impacts of Synchrophasor-Based DTR and SIPS Cyber–Physical Systems”, IEEE Systems Journal]. However, the cyber layer has never been considered in the proposed model, which I think is a lesser representation of the actual situation. Hence, it is suggested that the authors additionally consider the cyber layer modelling in their proposed model. Otherwise, they should provide a qualitative discussion on how the cyber layer could have been included and its potential impacts towards the presented results.

Reviewer #2: This manuscript proposes a Dynamics and resiliency of networks with concurrent cascading failure and self-healing. The idea is good and theoretically verified; however, it needs to spot the light on the main contribution, and many issues should be addressed according to the following comments:

1- The "Abstract" section should be more intensively focused on the main idea directly and must contain the contribution of this manuscript with numerical result indicators.

2-It is mandatory to add the figure of the system model.

check all the citing references of equations. In addition, check carefully all the abbreviation definitions, symbols, and standard units in the whole manuscript. I catch some errors and the other symbols were not defined.

3- The resolution and quality of all result figures should be modified; they should be presented as close to the camera-ready format. Also, please don't use the symbol abbreviations on X-Y-axes, they must have the full name with their units. Further, a lot of figures with poor quality X-Y axes, it must write by zooming in on the effective zones.

4- Table of comparison need to be addresses in various terms to highlight the performance of conducted work and related works in literature.

6. PLOS authors have the option to publish the peer review history of their article (what does this mean?). If published, this will include your full peer review and any attached files.

Reviewer #1: No

Reviewer #2: No

---

## [Editor Report · Decision Letter 1]

28 Oct 2022

Dynamics and resiliency of networks with concurrent cascading failure and self-healing

PONE-D-22-20047R1

Dear Dr. Zhang,

We’re pleased to inform you that your manuscript has been judged scientifically suitable for publication and will be formally accepted for publication once it meets all outstanding technical requirements.

Kind regards,

Sathishkumar V E

Academic Editor

PLOS ONE
---

## [Editor Report · Acceptance letter]

4 Nov 2022

PONE-D-22-20047R1 

Dynamics and resiliency of networks with concurrent cascading failure and self-healing 

Dear Dr. Zhang:

I'm pleased to inform you that your manuscript has been deemed suitable for publication in PLOS ONE. Congratulations! Your manuscript is now with our production department. 

Kind regards, 

on behalf of

Dr. Sathishkumar V E 

Academic Editor

PLOS ONE